# Housing structure including the surrounding environment as a risk factor for visceral leishmaniasis transmission in Nepal

Lina Ghassan Younis[1], Axel Kroeger[2,3], Anand B. Joshi[1], Murari Lal Das[1], Mazin Omer◉[1], Vivek Kumar Singh◉[1], Chitra Kumar Gurung[1], Megha Raj Banjara◉[4]*

1 Public Health and Infectious Disease Research Center, New Baneshwor, Kathmandu, Nepal, 2 University of Freiburg, Freiburg, Germany, 3 WHO Special Programme for Research and Training in Tropical Diseases (WHO-TDR), Geneva, Switzerland, 4 Central Department of Microbiology, Tribhuvan University, Kirtipur, Kathmandu, Nepal

* banjaramr@gmail.com

**Data Availability Statement:** All relevant data are within the manuscript and its supporting information files.

## Abstract

Visceral leishmaniasis (VL) in Nepal is found in 61 out of 75 districts including areas previously listed as non- endemic. This study focused on the role of housing conditions and its immediate environment in VL transmission, to limit future transmissions, ensure sustainable vector control and support the VL elimination program. The objective was to explore the risk factors in rural housing-and land lot typologies contributing to clinical VL occurrence and transmission. Housing structures and land lots were examined based on characteristics as risk factors of VL transmission in a case-control analysis. VL cases from 2013–2017 were identified based on the existing database from the Epidemiology and Disease Control Division and District Public Health Office from the plain Terai area (Morang, and Saptari districts) and hilly area (Palpa district) of Nepal. Two hundred and three built environments were analyzed (66 cases and 137 controls). Inferential statistics and logistic regression analysis were performed to determine the association of risk factors with VL. The risk factors with the highest odds of VL were: bamboo walls (adjusted odds ratio (AOR)- 8.1, 95% CI 2.40–27.63, $p$ = 0.001), walls made of leaves/branches (AOR- 3.0, 95% CI 0.84–10.93, $p$ = 0.090), cracks in bedroom walls (AOR- 2.9, 95% CI 0.93–9.19, $p$ = 0.065), and placing sacks near sleeping areas (AOR- 19.2, 95% CI 4.06–90.46, $p$ <0.001). Significant outdoor factors were: lots with Kadam trees (AOR- 12.7, 95% CI 3.28–49.09, $p$ <0.001), open ground-outdoor toilets (AOR- 9.3, 95% CI 2.14–369.85, $p$ = 0.003), moisture in outdoor toilet sheds (AOR- 18.09, 95% CI 7.25–451.01, $p$ = 0.002), nearby- open land (AOR- 36.8, 95% CI 3.14–430.98, $p$ = 0.004), moisture inside animal sheds (AOR- 6.9, 95% CI 1.82–26.66, $p$ = 0.005), and surrounding animals/animals wastes particularly goats (AOR- 3.5, 95% CI 1.09–10.94, $p$ = 0.036). Certain housing and surrounding environmental conditions and characteristics are risk factors for VL. Hence, elimination and educational programs should include the focus on housing improvement and avoidance of risk factors. Longitudinal interventional studies are required to document temporal relationships and whether interventions on these factors will have an impact on *Leishmania* transmission or burden.

**Funding:** The authors received no specific funding for this work.

**Competing interests:** The authors have declared that no competing interests exist.

## Author summary

Visceral leishmaniasis (VL) is a fatal disease if not treated in time. It is the disease of the poorest people. Poor housing and sanitation around the house are considered as the risk factors for the occurrence of VL. The main objective of our research was to explore multiple risk factors from different housing and land lot- typologies for VL occurrence and transmission through a case-control study. Some of our results reveal that inside the dwelling the likelihood of having clinical VL was substantially enhanced by providing suitable breeding sites for the insect vectors; as an example: the probability of getting VL disease was about ten times higher when there were sacks (empty or filled) near sleeping areas. Also cracks in walls and floors were found to contribute to vector transmission but also particular wall structures such as bamboo walls, and finishes such as animal manure were shown to be optimal for vector breeding. Furthermore, certain animals and plant types in the immediate environment seem to attract the vectors and to have a substantial effect on VL occurrence. Thus, given the alarming increase of VL in previously non-endemic areas of Nepal, these findings, among others, will allow readers and policymakers to better understand the "hidden" VL transmission factors, and will–hopefully- encourage initiating future studies.

## Introduction

Visceral leishmaniasis (VL) is considered one of the largest parasitic killers in the world after malaria [1], responsible for the global disease burden of 50,000 to 90,000 cases each year [2].

The Indian sub-continent alone has the biggest share with more than 60% of cases reported worldwide [3,4,5], peaking at the age of 12 years and younger [6,7]. Visceral leishmaniasis (known as kala-azar in the Sub-Indian continent), is characterized by high fever, enlargement of spleen and liver, muscle atrophy, and anaemia. It is often fatal without proper treatment [2,8,9].

This disease is also known as the disease of the poorest of the poor [10, 11], and 25–75% of households of VL patients experience financial crisis [8,12] because they are unable to work, and affected children stop attending schools.

Poverty and illiteracy are related to poor living conditions. Correspondingly, poor living conditions are proven to be strongly influencing the risk of VL [10]. Thus, level of exposure to surrounding nature and the construction of the residence (using local and affordable building materials and most importantly the way those materials are refined and finished) are closely related to human health which is the overall theme of this study.

VL caused by *Leishmania donovani* on the Indian sub-continent is transmitted by the bite of female *Phlebotomus argentipes* [13]. Until recently they were thought to be resting, breeding, and feeding only indoors and that they move short distances in a "hopping" way. Recent observations, however, have shown that some sand fly species are able to thrive outdoors and disperse in village vegetation such as banana and palm trees [14].

There are ecological parameters that influence oviposition and larval survival and development including rainfall, relative humidity, air temperature, wind speed, soil PH and moisture, and organic carbon [15], all of which are therefore related to climate change [16,17,18,19,20,21]. The ideal temperature for vector breeding of *P. argentipes* is 20–24.9˚C. Under these climatic conditions, eggs will hatch to larva within one to two weeks provided that moisture is present whether from rainfall, saturated soil, roof/wall/floor cracks, loose

bricks, animal burrows, and cattle sheds especially in rudimentary constructions such as from bamboo and mud [15].

The study was conducted at a point in time when the VL elimination program, which started in 2005, has reached in 2017 in Nepal its goal of less than one case per 10 000 population [22] but is now facing the spread of the disease to former non-endemic areas and the occasional appearance of clinically manifest disease in formerly endemic and non-endemic areas [23]. Most likely the spread of the disease is fuelled by climate change as the insect vectors are moving to higher altitudes and valleys in hilly districts as has been observed in Bangladesh [24]. Therefore, it is particularly important for the maintenance of the elimination success to identify risk factors for disease transmission in order to introduce preventive measures against the spread of the disease.

This study compares the housing conditions of previous VL cases and those without VL to identify possible risk factors for parasite transmission. The results could help to define housing conditions that could prevent the spread of VL.

## Methods

### Ethics statement

The research protocol was approved by the ethical committee of Nepal Health Research Council (NHRC). Before the interview of the household head and observation of the house, written consent was obtained. All of the household heads were adults.

### Study area

The study was conducted in three highly VL endemic districts Palpa, Morang and Saptari. Palpa district (province 5), where VL has been recently introduced, is located in the northern district of Nepal. It is a mountainous terrain, located about 324 Km from the Indian border and approximately at the same distance from Kathmandu. The district covers 1370 Km$^2$ of land and is populated by 261,180 inhabitants. Most villages in the district are below the poverty line with the majority of houses and surrounding plots made of mud and stone. The grassy ground has some plants, mainly flowers, banana, and palm trees. Farm animals live within the house surroundings in cattle sheds indoors or outdoors. Morang (province 1) and Saptari (province 2), on the other hand, are considered VL endemic districts with the biggest share of VL cases from within the last five years. The districts are situated in the Terai area, a highly agricultural region [25] both being 10 Km away from the Indian border, with an area of 1,360 Km$^2$ and 1,855 Km$^2$, an approximate population of 965,370 and 639,284 respectively. Especially in Saptari; many villages are far below the poverty line [16]. The majority of houses and surrounding plots are made of bamboo, leaves, and straw, with muddy ground. The surroundings are with some plants mainly Kadam, flowers, bananas, and corn. Farm animals are also within the compounds indoors or outdoors.

### Study design

We used quantitative method which included a case-control study with a formal interview survey and observations and in-depth interviews with the inhabitants. Cases were houses with clinical VL patients (i.e. excluding the probably large number of unknown sub-clinical cases) during the preceding five years and controls were houses without clinical VL patients during the preceding five years. The five-year period was chosen to include a sufficient large number of affected houses (as in recent years the case numbers were very low) and at the same time to

get a reflection of the current housing conditions (as most families live for several years in the same house).

## Data collection procedures

We calculated the sample size based on case-control study design using 95% CI, 80% power and estimated odds ratio of 2. Data collection was performed during the monsoon season (June/July 2018) in 203 households, 66 of which were cases and 137 controls. Lists of VL patients of five years (2013–2017) were obtained from the respective District Public/Health Office. It can be assumed that although there is a considerable diagnostic delay (from onset of symptoms until diagnosis of VL. [26], almost all patients appear at some stage at a public hospital where the diagnosis is established. Among the 66 VL cases, the last one occurred almost two months before the study period. Among these 66 households from three districts, 12 had more than one VL case in the preceding 5 years especially in Morang and Saptari. But only the most recent VL case was taken as reference for the study. Case houses were identified with the help of Female Community Health Volunteers. Control houses (two control houses per each case house) within a radius of 100 m of case houses were selected randomly using the lottery method.

Candidate risk factors for VL infection were related to i) *opportunities for vector breeding*: Construction age of the house, construction materials of walls, floors, and roofs, cracks in the wall, water pooling floor (unlevelled floors causing water to accumulate) and/or observed moisture inside the house particularly in the bedroom or in the cattle shed; indicators of untidiness (sacks, water bottles, animal waste, food in the rooms) ii) *opportunities for vector biting*; crowding conditions (number of inhabitants, number and area of roofed rooms), poor bedroom ventilation (i.e. low number of window openings and observed ventilation pattern), small bedroom size (small: $3m^2$ or big: more than $3m^2$), outdoor toilets and/or water sources obliging people to get out at night during the main biting times of the vector [12]; iii) *possible attractants for vectors*: Animal sheds (location and distance from household) and number and species of animals; certain plants in the environment.

## Data management and analysis

Data were transferred to the computer and SPSS was used for the analysis. The data analysis included the calculation of crude odds ratios (OR) and adjusted odds ratios (AOR) using bivariate and multiple logistic regression analyses. Unconditional logistic regression analysis was performed. The variables which had p-value of less than 0.05 in the bivariate logistic regression analysis (Tables 1, 2 and 3) were included in the multiple logistic regression analysis. In Table 4, only the significant variables (p-value < 0.05) in multiple logistic regression have been presented.

## Results

### Housing and settlement patterns

The bivariate analysis showed a number of significant housing risks for getting VL (Tables 1–3). Regarding risk factors related to improved opportunities for vector breeding, it was evident that natural floors were associated with an about eight times increased risk of VL as compared to any other material on the floor (cement, stones, etc.) (*p* = 0.039). Likewise, 50 years or older house constructions were three times likely to add more risk of VL occurrence, along with walls made from straw, leaves, and/or bamboo. Also, the indicators of untidiness (water/food or sacks in the bedroom) and cracks and/or moisture in walls/floors were significant risk

**Table 1. Bivariate analysis of housing conditions with the occurrence of VL.**

| Characteristics | Control (%) | Case (%) | Crude OR | CI 95% | P-Value |
|---|---|---|---|---|---|
| **House age in years** | | | | | |
| **1–10** | 77 (58.3) | 29 (45.3) | Ref | | |
| **11–50** | 50 (37.9) | 28 (43.8) | 1.487 | 0.79–2.79 | 0.217 |
| **51 or more** | 5 (3.8) | 7 (10.9) | 3.717 | 1.09–12.65 | 0.036 |
| **Number of windows/openings** | | | | | |
| **0–3** | 46 (37.4) | 36 (56.3) | Ref | | 0.124 |
| **8 or more** | 34 (27.6) | 8 (12.5) | 0.405 | 0.11–1.56 | 0.189 |
| **4–7** | 43 (35) | 20 (31.3) | 0.565 | 0.31–1.05 | 0.069 |
| **Walls** | | | | | |
| **Stone wall** | 9 (6.6) | 4 (6.1) | 0.918 | 0.27–3.10 | 0.890 |
| **Straw wall** | 39 (28.5) | 41 (62.1) | 4.121 | 2.22–7.67 | <0.001 |
| **Leaves wall** | 66 (48.2) | 49 (74.2) | 3.101 | 1.63–5.91 | 0.001 |
| **Wooden wall** | 7 (5.1) | 2 (3.1) | 0.590 | 0.12–2.92 | 0.517 |
| **Brick wall** | 24 (17.5) | 10 (15.2) | 0.841 | 0.38–1.88 | 0.673 |
| **Cement/dried mud wall** | 30 (21.9) | 3 (4.5) | 0.170 | 0.05–0.58 | 0.005 |
| **Bamboo wall** | 30 (21.9) | 38 (57.6) | 4.840 | 2.57–9.13 | <0.001 |
| **Floors** | | | | | |
| **Stone/tile floor** | 3 (2.2) | 0 | 0.000 | 0.00–0 | 0.99 |
| **Natural floor** | 121 (88.3) | 65 (98.5) | 8.595 | 1.12–66.28 | 0.039 |
| **Wood floor** | 4 (2.9) | 1 (1.5) | 0.512 | 0.06–4.67 | 0.552 |
| **Roof** | | | | | |
| **Asbestos roof** | 7 (5.1) | 0 | 0.000 | 0.00–0 | 0.99 |
| **Straw roof** | 29 (21.2) | 16 (24.2) | 1.192 | 0.59–2.39 | 0.622 |
| **Leave roof** | 18 (13.1) | 15 (22.7) | 1.944 | 0.91–4.16 | 0.086 |
| **Wooden roof** | 31 (22.6) | 12 (18.2) | 0.760 | 0.36–1.60 | 0.469 |
| **Brick roof** | 6 (4.4) | 0 | 0.000 | 0.00–0 | 0.99 |
| **Cement roof** | 9 (6.6) | 2 (3.0) | 0.444 | 0.09–2.12 | 0.309 |
| **Corrugated roof** | 115 (83.9) | 56 (84.8) | 1.071 | 0.48–2.42 | 0.868 |
| **Bamboo roof** | 93 (67.9) | 51 (77.3) | 1.609 | 0.82–3.17 | 0.170 |
| **Moisture** | | | | | |

*(Continued)*

**Table 1.** (Continued)

| Characteristics | Control (%) | Case (%) | Crude OR | CI 95% | P-Value |
|---|---|---|---|---|---|
| Kitchen | 52 (38.0) | 39 (60.0) | 2.452 | 1.34–4.49 | 0.004 |
| Bedroom | 53 (38.7) | 43 (66.2) | 3.098 | 1.67–5.75 | <0.001 |
| Toilet | 43 (31.4) | 40 (61.5) | 3.498 | 1.89–6.48 | <0.001 |
| Living room | 50 (36.5) | 39 (60.0) | 2.610 | 1.42–4.78 | 0.002 |
| Presence of shed | 44 (32.1) | 40 (61.5) | 3.382 | 1.88–6.26 | <0.001 |
| Court in the house | 73 (53.3) | 45 (69.2) | 1.973 | 1.06–3.68 | 0.033 |
| Presence of cracks | 66 (50.4) | 43 (67.2) | 2.017 | 1.08–3.76 | 0.028 |
| Indoor stored water material | | | | | |
| Metal | 80 (59.3) | 50 (75.8) | Ref | | 0.161 |
| Earth | 6 (4.4) | 0 | 0 | 0 | 0.99 |
| Plastic | 49 (36.3) | 16 (24.2) | 0.522 | 0.27–1.02 | 0.056 |
| Ventilation | | | | | |
| Ventilation kitchen | 78 (58.6) | 33 (53.2) | 0.802 | 0.44–1.47 | 0.477 |
| Ventilation bedroom | 66 (62.3) | 37 (69.8) | 1.402 | 0.69–2.84 | 0.349 |
| Toilet location | | | | | |
| None | 60 (43.8) | 40 (60.6) | Ref | | 0.092 |
| Inside | 1 (0.7) | 0 | 0 | 0 | 1 |
| Outside | 76 (55.5) | 26 (39.4) | 0.513 | 0.28–0.93 | 0.029 |
| No Spraying intervention | 66 (66.7) | 22 (41.5) | 2.818 | 1.42–5.61 | 0.003 |
| Food in room | 18 (14.0) | 20 (32.3) | 2.937 | 1.42–6.09 | 0.004 |
| Water in room | 2 (1.6) | 5 (8.1) | 5.570 | 1.05–29.57 | 0.044 |
| Sacks in room | 4 (3.1) | 16 (25.8) | 10.870 | 3.45–34.21 | <0.001 |
| Animals/ animal waste in room | 3 (2.3) | 4 (6.5) | 2.897 | 0.63–13.36 | 0.173 |

factors for VL. Particularly households with sacks indoors had a ten times higher risk of occurrence of VL than households without sacks indoors (OR- 10.87, $p$ <0.001), Other risk factors included moisture in the kitchen, toilet, living spaces, the presence of courtyards with odds ratios 2.45, 3.50, 2.61, 1.97 respectively.

Animals, particularly cattle, are attractants for vectors and the floor and dung are preferred breeding places; therefore, as expected, the shed location (especially in-front and a side of the house/sleeping area) was found to increase approximately four times the probability of disease

Table 2. Bivariate analysis of animals, animal shed, and plants around the house as VL risk factors.

| Characteristics | Control (%) | Case (%) | Crude OR | CI 95% | P-Value |
|---|---|---|---|---|---|
| **Animals in the house** | | | | | |
| Cow | 71 (51.8) | 46 (69.7) | 2.138 | 1.15–3.99 | 0.017 |
| Buffalo | 38 (27.7) | 21 (31.8) | 1.216 | 0.64–2.30 | 0.549 |
| Ox | 11 (8.0) | 2 (3.0) | 0.358 | 0.08–1.66 | 0.190 |
| Pig | 17 (12.4) | 9 (13.6) | 1.115 | 0.47–2.65 | 0.806 |
| Dog | 16 (11.7) | 8 (12.1) | 1.043 | 0.42–2.58 | 0.927 |
| Duck | 6 (4.4) | 2 (3.0) | 0.682 | 0.13–3.48 | 0.645 |
| Chicken | 39 (28.5) | 25 (37.9) | 1.532 | 0.82–2.85 | 0.178 |
| Horse | 0 | 1 (1.5) | 0.000 | 0.00–0 | 0.99 |
| Goat | 83 (60.6) | 49 (74.2) | 1.875 | 0.98–3.59 | 0.058 |
| **Animal- shed distance** | | | | | |
| 80m or more | 23 (18.0) | 3 (4.8) | Ref | | 0.066 |
| 1-20m | 101 (78.9) | 56 (90.3) | 4.251 | 1.22–14.79 | 0.023 |
| 21-80m | 4 (3.1) | 3 (4.8) | 5.750 | 0.84–39.24 | 0.074 |
| **Animal number** | | | | | |
| 0–5 | 75 (55.1) | 29 (45.3) | Ref | | 0.191 |
| 6–12 | 41 (30.1) | 19 (29.7) | 1.198 | 0.60–2.40 | 0.608 |
| 13 or more | 20 (14.7) | 16 (25.0) | 2.069 | 0.94–4.53 | 0.069 |
| **Animal-shed location** | | | | | |
| None | 18 (22.0) | 4 (7.0) | Ref | | 0.157 |
| Front | 32 (39.0) | 32 (56.1) | 4.500 | 1.37–14.78 | 0.013 |
| Side | 21 (25.6) | 18 (31.6) | 3.857 | 1.10–13.50 | 0.035 |
| Back | 6 (7.3) | 3 (5.3) | 2.250 | .39–13.07 | 0.366 |
| Under house | 5 (6.1) | 0 | 0.000 | 0.00–0 | 0.99 |
| **Plants*** | | | | | |
| Mango | 55 (40.7) | 26 (40.0) | 0.970 | 0.53–1.77 | 0.920 |
| Corn | 8 (5.9) | 4 (6.1) | 1.024 | .30–3.35 | 0.970 |
| Cucumber/vegetables | 5 (3.7) | 3 (4.5) | 1.238 | 0.29–5.35 | 0.775 |
| Boga tree | 4 (3.0) | 3 (4.5) | 1.560 | 0.34–7.18 | 0.568 |

*(Continued)*

**Table 2.** (Continued)

| Characteristics | Control (%) | Case (%) | Crude OR | CI 95% | P-Value |
|---|---|---|---|---|---|
| Lychee | 8 (5.9) | 5 (7.6) | 1.301 | 0.41–4.14 | 0.656 |
| Tar | 12 (8.9) | 10 (15.2) | 1.830 | 0.75–4.49 | 0.186 |
| Guava | 16 (11.9) | 7 (10.6) | 0.882 | 0.34–2.26 | 0.795 |
| Banana | 20 (14.8) | 11 (16.7) | 1.150 | 0.52–2.57 | 0.733 |
| Lemon | 4 (3.0) | 2 (3.0) | 1.023 | 0.18–5.74 | 0.979 |
| Guard | 12 (8.9) | 5 (7.6) | 0.840 | 0.28–2.49 | 0.754 |
| Pumpkin | 15 (11.1) | 10 (15.2) | 1.429 | 0.60–3.38 | 0.417 |
| Kadam | **30 (22.2)** | **30 (45.5)** | **2.917** | **1.55–5.49** | **0.001** |
| Cactus | 1 (0.7) | 2 (3.0) | 4.187 | 0.37–47.04 | 0.246 |
| Jack fruit | 17 (12.6) | 9 (13.6) | 1.096 | 0.46–2.61 | 0.836 |
| Grains/nuts | 3 (2.2) | 1 (1.5) | 0.677 | 0.07–6.64 | 0.738 |
| Coconut | 17 (12.6) | 4 (6.1) | 0.448 | 0.14–1.39 | 0.164 |
| Pomegranate | 1 (0.7) | 2 (3.0) | 4.187 | 0.37–47.04 | 0.246 |
| Bamboo | 20 (14.8) | 9 (13.6) | 0.908 | 0.39–2.12 | 0.823 |
| Papaya | 7 (5.2) | 3 (4.5) | 0.874 | 0.22–3.48 | 0.845 |
| Flower | 35 (25.9) | 15 (22.7) | 0.840 | 0.42–1.68 | 0.622 |

*Common, non-scientific plant names.

occurrence OR- 4.50, $p$ = 0.013 (front), and OR- 3.857, $p$ = 0.035 (side). In the same way, the presence of cows and goats in the house, were found to be significant risk factors for the occurrence of VL. According to local beliefs plants around the house may attract sand fly vectors, however, only Kadam trees seem to be related to increased VL occurrence (Table 2).

It was found that land lots without any spraying intervention (inside or outside) were a higher risk for VL occurrence than with it (OR- 2.818, $p$ = 0.003). Grassland covers may be considered as preventive measures instead of muddy ones. The VL risk was about eighteen times higher when there was an open field/land near the sleeping area (OR- 18.52, $p$ <0.001) (Table 3).

The multivariate analyses leading to adjusted odds ratio summarized the most prominent housing and environment-related risk factors for VL transmission (i.e. household/environmental characteristics significantly associated with VL or non-VL in a household; Table 4).

*Characteristics of walls*: Houses with walls made of leaves/branches (AOR- 3.0, p = 0.090), or of bamboo (AOR- 8.1, p = 0.001) had a high risk for VL occurrence. Walls with cracks especially in the bedroom had a three times higher risk of VL occurrence than properly- constructed/ finished walls.

Table 3.  **Bivariate analysis of characteristics around the house as risk factors of VL.**

| Characteristics | Control (%) | Case (%) | Crude Ratio OR | CI 95% | P-Value |
|---|---|---|---|---|---|
| **Rice field near room** | 0 | 6 (9.7) | 0 | 0 | 1 |
| **Open land near room** | **2 (1.6)** | **14 (22. 6)** | **18.521** | **4.06–84.54** | **<0.001** |
| **Pond near room** | 0 | 4 (6.5) | 0 | 0 | 1 |
| **Toilet type(open ground)** | **85 (63.4)** | **51 (79.7)** | **2.262** | **1.12–4.57** | **0.023** |
| **Surrounding grounds** | | | | | |
| **Muddy area** | 125 (91.9) | 61 (95.3) | 1.789 | 0.48–6.65 | 0.385 |
| **Rocky area** | 14 (10.3) | 3 (4.7) | 0.429 | 0.12–1.55 | 0.196 |
| **Grass area** | **59 (43.4)** | **12 (18.8)** | **3.320** | **1.63–6.78** | **0.001** |
| **Distance to the water source** | | | | | |
| **21-80m** | 22 (16.5) | 14 (23.7) | Ref | | 0.432 |
| **80m or more** | 18 (13.5) | 9 (15.3) | 0.786 | 0.28–2.23 | 0.651 |
| **1-20m** | 93 (69.9) | 36 (61.0) | 0.608 | 0.28–1.32 | 0.207 |

Table 4.  **Multivariate analysis of risk factors for VL.**

| Risk Factors | Control (%) | Case (%) | Adjusted Odds Ratio- AOR | CI 95% of AOR | P-Value |
|---|---|---|---|---|---|
| **Characteristics of Walls** | | | | | |
| **Wall made of leaves** | 66 (48.2) | 49 (74.2) | 3.03 | 0.84–10.93 | 0.090 |
| **Bamboo wall** | 30 (21.9) | 38 (57.6) | 8.1 | 2.39–27.63 | 0.001 |
| **Cracks in walls** | 66 (50.4) | 43 (67.2) | 2.9 | 0.93–9.19 | 0.065 |
| **Toilet outside the house** | 43 (31.4) | 40 (61.5) | 18.09 | 7.25–451.9 | 0.002 |
| **Toilet type(open ground)** | 85 (63.4) | 51 (79.7) | 9.2 | 2.14–369.8 | 0.003 |
| **Animals** | | | | | |
| **Indoor animal shed** | 44 (32.1) | 40 (61.5) | 6.96 | 1.82–26.66 | 0.005 |
| **Goat** | 83 (60.6) | 49 (74.2) | 3.4 | 1.08–10.94 | 0.036 |
| **Plants** | | | | | |
| **Kadam** | 30 (22.2) | 30 (45.5) | 12.7 | 3.28–49.09 | <0.001 |
| **Household items** | | | | | |
| **Sacks in room** | 4 (3.1) | 16 (25.8) | 19.2 | 4.05–90.45 | <0.001 |
| **Open lands/fields near sleeping area** | 2 (1.6) | 14 (22. 6) | 36.8 | 3.14–430.9 | 0.004 |

*Household untidiness*: Empty or filled sacks containing different items in the bedroom as indicators of untidiness were risk factors for VL occurrence (AOR- 19.2, $p$ <0.001).*Toilet location*: Outdoor toilets and particularly open ground-outdoor toilets were found to be a significant risk factor for VL occurrence: AOR- 18.09, $p$ = 0.002 and AOR- 9.3, $p$ = 0.003 respectively.

*Characteristics of the surrounding area (about 50m around houses)*: Open land areas near sleeping areas were found to be a significant risk factor (AOR- 36.8, $p$ = 0.004).

*Animals*: Animal sheds (AOR- 6.9, $p$ = 0.005) and animals in the house (particularly goats) (AOR- 3.5, $p$ = 0.036) were found to be risk factors for VL occurrence.

*Plants*: Kadam trees around the house seem to increase the risk of VL; AOR- 12.7, p<0.001.

## Discussion

The importance of housing conditions for the health of the inhabitants has gained increasing attention (e.g. WHO 2018 [27]). Our study has shown evidence on the impact of housing factors including the surrounding environment on the occurrence of clinical VL in its inhabitants. Particularly important are the construction materials which provide either shelter to the insect vectors or suitable breeding places [10]. They are routinely applied in rural Nepal, such as the use of animal waste -for example, cow/buffalo waste or "Gobar"- not only as fertilizer but also for wall and floor plastering as easily available, no-cost materials; they were found to be risk factors for the occurrence of VL most probably by favoring indoor sand fly breeding and the movement of the vectors on the wall.

The recommendation for house construction or renovation would be, to apply these dung materials evenly, finished and closed, in addition, to being vertically-leveled in order to minimize or eliminate vector breeding and hiding places.

In some lots, local dwellers used bamboo as the main structure, which is sustainable. However, if left untreated, it will lose its natural durability and will shrink creating structural holes. Furthermore, bamboo at a certain age has higher sugar content which is considered to be a good food source for insects, thus, the risk of insect infestation will be much higher [28]. Our study showed bamboo construction to be an important risk factor for VL transmission which should be addressed in educational programs.

Different studies showed evidence on the relationship between the location of openings (windows/doors) and vector movement [29], but very few analyzed other factors such as the construction age of houses, which has an important association with vector breeding [30]. In our study old houses (constructions more than 50 years old), were an important risk factor for VL as the age of the building is linked to other factors such as having cracks in walls/floors, infested roofs and walls, uneven-humped floors.

Household untidiness was also an association with increased VL transmission. In houses where the bedroom was used as a storage place for sacks and many other items, sandfly vectors apparently found easy breeding and hiding places.

Outdoor toilets, particularly open-ground toilets, are usually surrounded by a wet floor as an ideal breeding place for sand flies; additionally, people who go at night during the main vector biting time to the toilet may be exposed to infective bites.

Better known risk factors for VL transmission are the presence of animal sheds close to houses and bedroom [15] which was confirmed by our study. The mere presence of yards in the compound or open land areas near the bedroom as ideal vector breeding places also increased the VL risk substantially. The importance of village vegetation for the breeding and survival of sand fly vectors has already been established by a previous study [14]. In this study, we found the presence of "Kadam trees"- to be significantly associated with VL disease. The

challenging part, however, is minimizing the associated risks of Kadam trees while they are being known to have many benefits and medicinal properties and are sacredly planted near temples [31, 32]. More studies are required to confirm the role of specific trees (such as Kadam trees) and animals (such as goats) in vector transmission.

The limitations of our study were not only the case-control design but include also the difficulty of establishing the causality of risk factors as well as the interdependency of many factors; longitudinal and interventional studies are required to document causal relationships. We calculated the sample size taking into account only one explanatory variable, therefore the statistical power for detecting significant associations could be lower when analyzing multiple variables simultaneously. One particular challenge in VL field studies is the unknown number of asymptomatic cases [33]. However, as the ratio between clinical VL cases and asymptomatic cases is fairly constant, we can take the socio-economically more important clinical VL cases as an indicator of VL transmission.

In general terms, this and related studies on preventive measures are important for the maintenance of the VL elimination achievements [33, 34] which are not only threatened by the decrease of Public Health activities after reaching the elimination goal (abandoning vector control and early case detection and treatment) but also the challenges of environmental and climate change which contribute to the spread of VL [16].

## Conclusions

The community should be informed about the importance of housing conditions as VL risk factors (bamboo walls, cracks/ water pooling inside and outside the house and around animal shed should be avoided). Animal manure should immediately be collected, cleaned and dried properly to be used for cooking or wall plastering (complete and leveled plastering of the whole wall is an essential requirement). Hygiene of the living/sleeping area is important (no food, open water sources, nor filled and empty sacks). The importance of sleeping indoors and having screened windows and well-structured/ finished walls for vector control should be communicated to the community. Frontline health workers should provide proper information messages in the communities regarding VL transmission, house and environment as risk factors, and the need for protective measures.

## Ethics approval and consent to participate

The research protocol was approved by the ethical committee of Nepal Health Research Council (NHRC). Before the interview of the household head and observation of the house, written consent was obtained.

## Supporting information

**S1 Checklist. STROBE checklist.**
(DOC)

## Acknowledgments

We would like to thank Mr. Krishna Raj Pant for the management of data collection. We would like to thank Prof. Greg Matlashewski from McGill University, Canada for editing the manuscript.

## Author Contributions

**Conceptualization:** Lina Ghassan Younis, Axel Kroeger, Anand B. Joshi, Megha Raj Banjara.

**Data curation:** Lina Ghassan Younis, Anand B. Joshi, Murari Lal Das, Mazin Omer, Vivek Kumar Singh, Chitra Kumar Gurung, Megha Raj Banjara.

**Formal analysis:** Lina Ghassan Younis, Anand B. Joshi, Chitra Kumar Gurung, Megha Raj Banjara.

**Investigation:** Lina Ghassan Younis, Mazin Omer.

**Methodology:** Lina Ghassan Younis, Axel Kroeger, Anand B. Joshi, Murari Lal Das.

**Project administration:** Lina Ghassan Younis, Mazin Omer, Vivek Kumar Singh, Chitra Kumar Gurung.

**Supervision:** Anand B. Joshi, Murari Lal Das, Chitra Kumar Gurung, Megha Raj Banjara.

**Writing – original draft:** Lina Ghassan Younis, Megha Raj Banjara.

**Writing – review & editing:** Axel Kroeger, Anand B. Joshi, Murari Lal Das, Mazin Omer, Vivek Kumar Singh, Chitra Kumar Gurung, Megha Raj Banjara.

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
