## [Decision Letter · Decision Letter 0]

9 Dec 2019

Dear Dr Banjara:

Thank you very much for submitting your manuscript "Housing structure including the surrounding environment as a risk factor for visceral leishmaniasis transmission in Nepal" (#PNTD-D-19-01743) for review by PLOS Neglected Tropical Diseases. Your manuscript was fully evaluated at the editorial level and by independent peer reviewers. The reviewers appreciated the attention to an important problem, but raised some substantial concerns about the manuscript as it currently stands. These issues must be addressed before we would be willing to consider a revised version of your study. We cannot, of course, promise publication at that time.

We therefore ask you to modify the manuscript according to the review recommendations before we can consider your manuscript for acceptance. Your revisions should address the specific points made by each reviewer. 

When you are ready to resubmit, please be prepared to upload the following:

(1) A letter containing a detailed list of your responses to the review comments and a description of the changes you have made in the manuscript.

(2) Two versions of the manuscript: one with either highlights or tracked changes denoting where the text has been changed (uploaded as a "Revised Article with Changes Highlighted" file); the other a clean version (uploaded as the article file).

(3) If available, a striking still image (a new image if one is available or an existing one from within your manuscript). If your manuscript is accepted for publication, this image may be featured on our website. Images should ideally be high resolution, eye-catching, single panel images; where one is available, please use 'add file' at the time of resubmission and select 'striking image' as the file type. 

Please provide a short caption, including credits, uploaded as a separate "Other" file. If your image is from someone other than yourself, please ensure that the artist has read and agreed to the terms and conditions of the Creative Commons Attribution License at http://journals.plos.org/plosntds/s/content-license (NOTE: we cannot publish copyrighted images). 

(4) If applicable, we encourage you to add a list of accession numbers/ID numbers for genes and proteins mentioned in the text (these should be listed as a paragraph at the end of the manuscript). You can supply accession numbers for any database, so long as the database is publicly accessible and stable. Examples include LocusLink and SwissProt.

(5) To enhance the reproducibility of your results, we recommend that you deposit your laboratory protocols in protocols.io, where a protocol can be assigned its own identifier (DOI) such that it can be cited independently in the future. For instructions see http://journals.plos.org/plosntds/s/submission-guidelines#loc-methods

While revising your submission, please upload your figure files to the Preflight Analysis and Conversion Engine (PACE) digital diagnostic tool, https://pacev2.apexcovantage.com/ PACE helps ensure that figures meet PLOS requirements. To use PACE, you must first register as a user. Then, login and navigate to the UPLOAD tab, where you will find detailed instructions on how to use the tool. If you encounter any issues or have any questions when using PACE, please email us at figures@plos.org.

We hope to receive your revised manuscript by Feb 07 2020 11:59PM. If you anticipate any delay in its return, we ask that you let us know the expected resubmission date by replying to this email.

To submit a revision, go to https://www.editorialmanager.com/pntd/ and log in as an Author. You will see a menu item call Submission Needing Revision. You will find your submission record there. 

Sincerely,

Guilherme L Werneck

Associate Editor

Shan Lv

Deputy Editor

Reviewer's Responses to Questions

**Key Review Criteria Required for Acceptance?**

**Methods**

-Are the objectives of the study clearly articulated with a clear testable hypothesis stated?

-Is the study design appropriate to address the stated objectives?

-Is the population clearly described and appropriate for the hypothesis being tested?

-Is the sample size sufficient to ensure adequate power to address the hypothesis being tested?

-Were correct statistical analysis used to support conclusions?

-Are there concerns about ethical or regulatory requirements being met?

Reviewer #1: Methods seem well handled

Reviewer #2: (No Response)

Reviewer #3: This is an interesting paper approaching the housing structure and the nearby environment as factors associated with the risk of having a VL case in the house during the past five years. Cases (houses with VL cases) were obtained from the surveillance registry of VL cases and controls (houses without VL cases) were randomly selected in the neighborhood of the cases. Some information on the background of the study site and some methodological options used for exploring the study hypothesis were partially or no described as it follows:

1. The estimated underreporting of VL cases, considering that the registry of cases was made retrospectively;

2. How could the authors be sure that controls were free of VL cases during the five years period? And what was the rational for choosing a five years period?

3. A relatively high proportion of the exposures under exploration could be affected by collinearity issues. There is no mention to such a detail in the methods section, please clarify.

**Results**

-Does the analysis presented match the analysis plan?

-Are the results clearly and completely presented?

-Are the figures (Tables, Images) of sufficient quality for clarity?

Reviewer #1: Interpretation is poor and analysis requires attention

Reviewer #2: (No Response)

Reviewer #3: Results are clear and the exposures were divided into different categories, but there was lacking a rational framework organizing the approach. It seems to me that some exposures could be more proximal to a successful VL transmission case than others. Did the authors think the possibility of organizing an a priori explanatory model to be submitted to the hypothesis testing?

As we know, most of the VL infection cases remained asymptomatic. Then, houses without VL symptomatic VL cases could have asymptomatic infections. If it were the case, the whole methodological approach could be questioned. Please consider adding a paragraph on this issue in the discussion section.

**Conclusions**

-Are the conclusions supported by the data presented?

-Are the limitations of analysis clearly described?

-Do the authors discuss how these data can be helpful to advance our understanding of the topic under study?

-Is public health relevance addressed?

Reviewer #1: Needs revision

Reviewer #2: (No Response)

Reviewer #3: Most of the explanations offered in the discussion section are reasonable points but in fact the methodological approach does not allow definite conclusions on the role of the factors identified as associated to VL transmission. Then, the conclusions should be less emphatic on the practicalities, although all the proposed preventive measures must be implemented for other reasons beyond the VL risk, the scientific conclusions of the paper should be restricted to the facts with their respective limitations.

**Editorial and Data Presentation Modifications?**

Reviewer #1: Major revision

Reviewer #2: (No Response)

Reviewer #3: Line 86 – P. argentipes. Please put the genus name in full. 

Line 95. P. argentipes . Please change to the abbreviated genus name.

Line 100. Please consider rephrasing the sentence: “This study compares the housing conditions of previous VL cases and without VL”. 

Line 151: Please define immediate environment.

**Summary and General Comments**

Reviewer #1: The manuscript deals with an important issue of South Asia that affects huge number of people every year. The authors examined how environmental factors particularly housing structure implicate spread or incidence of VL in Nepal with a case study. Given that climate change and anthropogenic activities could instigate VL vectors into new locations, the study could be useful to develop targeted interventions to save life from VL. So many studies have done in South Asia to map and analyse VL occurrences and there is less room to add new knowledge. Also, there was an elimination program of VL which ran for a decade or more, reading this work gives me an impression that the elimination program possibly did not help much, at least in Nepal. Hence this work could be useful given that the following issues are addressed adequately. 

[1] Existing works and review of the VL in South Asia is poorly handled which shows that the motivation of this work is poor. Therefore there are a few quality works that could help restructuring of the introduction and discussion parts. I am suggesting a few below but more could be obtained from the net via search engine such as Google Scholar. 

https://tropmedhealth.biomedcentral.com/articles/10.1186/s41182-017-0054-9

https://www.sciencedirect.com/science/article/pii/S1473309910703200

https://www.bmj.com/content/364/bmj.k5224.abstract

https://tropmedhealth.biomedcentral.com/articles/10.1186/s41182-017-0069-2

https://journals.plos.org/plosntds/article?rev=2&id=10.1371/journal.pntd.0007724

https://onlinelibrary.wiley.com/doi/abs/10.1111/tbed.13416

https://parasitesandvectors.biomedcentral.com/articles/10.1186/s13071-019-3778-z

[2] Line 85-90: You are pointing out about environmental change but this part needs really fleshing out more as to how environmental change is affecting the spread of VL in Nepal in particular and in South Asia in general. This would could be useful though https://link.springer.com/chapter/10.1007/978-3-319-47101-3_19

[3] Line 96–99: good statements but requires more evidence, and above works could provide more evidence. 

[4] What is the point of highlighting Table 1? Should be in regular style 

[5] Discussion section: you should relate findings of yours with existing works, what similarities and dissimilarities and why? The results of elimination program also need be highlighted here to show why it worked and why not?

Reviewer #2: The present work evaluate variables associated to residences and the environment and relate them to the occurrence of visceral leishmaniasis cases in Nepal.

Despite its importance to local visceral leishmaniasis transmission, i chose to reject the work because i consider the paper do not attend the PLOS NTDs criteria for publication.

One important issue in the decision was the lack of innovation of the work, the application of a predictive model based on mathematicals tools that could influence local control programs and consequently the incidence of cases would strengthen the study. The work is conducted well , with adequate analysis but no novelty in the scientific context.

Some variables evaluated were not observed in the control group and the authors do not make clear whether they were used for multivariate analysis, this may also influence the results.

Another inadequacy is associated with the conclusion that it should propose control measures through the application of a predictive model and not just inform those exposed.

These are some examples that led me to the decision to reject the job. Implementing a predictive model or an extension of the analysis so that environmental findings can be correlated with personal data could strengthen the article in the future.

Reviewer #3: This is an original paper on a relevant public health issue. Authors approached an important set of exposures that are vulnerable to preventive interventions. The methodological approach is correct but deserves some improvements related to the description of the analytical procedures and the interpretation of the main results should be better contextualized, considering the study limitations. Please see specific comments above.

PLOS authors have the option to publish the peer review history of their article (what does this mean?). If published, this will include your full peer review and any attached files.

Reviewer #1: No

Reviewer #2: No

Reviewer #3: Yes: Gustavo Adolfo Sierra Romero

---

## [Decision Letter · Decision Letter 1]

11 Feb 2020

Dear Dr Banjara,

We are pleased to inform you that your manuscript 'Housing structure including the surrounding environment as a risk factor for visceral leishmaniasis transmission in Nepal' has been provisionally accepted for publication in PLOS Neglected Tropical Diseases.

Before your manuscript can be formally accepted you will need to complete some formatting changes, which you will receive in a follow up email. A member of our team will be in touch within two working days with a set of requests.

Best regards,

Guilherme L Werneck

Associate Editor

Shan Lv

Deputy Editor

Reviewer's Responses to Questions

**Key Review Criteria Required for Acceptance?**

**Methods**

-Are the objectives of the study clearly articulated with a clear testable hypothesis stated?

-Is the study design appropriate to address the stated objectives?

-Is the population clearly described and appropriate for the hypothesis being tested?

-Is the sample size sufficient to ensure adequate power to address the hypothesis being tested?

-Were correct statistical analysis used to support conclusions?

-Are there concerns about ethical or regulatory requirements being met?

Reviewer #1: Yes, I am fine with the revision. My concerns have now been addressed adequately.

Reviewer #3: The requested modifications/explanations were done.

**Results**

-Does the analysis presented match the analysis plan?

-Are the results clearly and completely presented?

-Are the figures (Tables, Images) of sufficient quality for clarity?

Reviewer #1: This section is OK

Reviewer #3: The requested modifications/explanations were done.

**Conclusions**

-Are the conclusions supported by the data presented?

-Are the limitations of analysis clearly described?

-Do the authors discuss how these data can be helpful to advance our understanding of the topic under study?

-Is public health relevance addressed?

Reviewer #1: Yes

Reviewer #3: The requested modifications/explanations were done.

**Editorial and Data Presentation Modifications?**

Reviewer #1: Accept

Reviewer #3: None

**Summary and General Comments**

Reviewer #1: The revision is OK with me

Reviewer #3: The current version of the paper is suitable for publication.

PLOS authors have the option to publish the peer review history of their article (what does this mean?). If published, this will include your full peer review and any attached files.

Reviewer #1: No

Reviewer #3: Yes: Gustavo Romero

---

## [Editor Report · Acceptance letter]

25 Feb 2020

Dear Dr. Banjara,

We are delighted to inform you that your manuscript, "Housing structure including the surrounding environment as a risk factor for visceral leishmaniasis transmission in Nepal," has been formally accepted for publication in PLOS Neglected Tropical Diseases.

Best regards,

Serap Aksoy

Editor-in-Chief

Shaden Kamhawi

Editor-in-Chief
